# National priority setting partnership using a Delphi consensus process to develop neonatal research questions suitable for practice-changing randomised trials in the United Kingdom

Katie Evans [1], Cheryl Battersby [1], James P Boardman [2], Elaine M Boyle,[3] William D Carroll,[4] Kate Dinwiddy,[5] Jon Dorling,[6] Katie Gallagher [7] Pollyanna Hardy,[8] Emma Johnston,[9] Helen Mactier [10] Claire Marcroft,[11] James Webbe [1], Chris Gale [1]

For numbered affiliations see end of article.

**Correspondence to**
Dr Chris Gale;
christopher.gale@imperial.ac.uk

## ABSTRACT

**Introduction** Methodologically robust clinical trials are required to improve neonatal care and reduce unwanted variations in practice. Previous neonatal research prioritisation processes have identified important research themes rather than specific research questions amenable to clinical trials. Practice-changing trials require well-defined research questions, commonly organised using the Population, Intervention, Comparison, Outcome (PICO) structure. By narrowing the scope of research priorities to those which can be answered in clinical trials and by involving a wide range of different stakeholders, we aim to provide a robust and transparent process to identify and prioritise research questions answerable within the National Healthcare System to inform future practice-changing clinical trials.

**Methods and analysis** A steering group comprising parents, doctors, nurses, allied health professionals, researchers and representatives from key organisations (Neonatal Society, British Association of Perinatal Medicine, Neonatal Nurses Association and Royal College of Paediatrics and Child Health) was identified to oversee this project. We will invite submissions of research questions formatted using the PICO structure from the following stakeholder groups using an online questionnaire: parents, patients, healthcare professionals and academic researchers. Unanswered, non-duplicate research questions will be entered into a three-round eDelphi survey of all stakeholder groups. Research questions will be ranked by mean aggregate scores.

**Ethics and dissemination** The final list of prioritised research questions will be disseminated through traditional academic channels, directly to key stakeholder groups through representative organisations and on social media. The outcome of the project will be shared with key research organisations such as the National Institute for Health Research. Research ethics committee approval is not required.

## STRENGTHS AND LIMITATIONS OF THIS STUDY

⇒ By involving parents and former patients alongside a wide range of healthcare professionals, we will ensure that research questions are important to all key stakeholder groups.

⇒ We will use established strategies (three-round eDelphi process) to rank research questions based on mean scores.

⇒ The study will rank research questions based on subjective input from a large number of key stakeholders; however, questions may need further work prior to being addressed in a clinical trial.

⇒ The final list of prioritised questions will be disseminated widely to inform neonatal clinical research.

## INTRODUCTION

The importance of involving different stakeholders including parents, patients, healthcare professionals and researchers to identify and prioritise research is well recognised.[1] In neonatal and perinatal medicine, these stakeholder groups have prioritised research uncertainties in the fields of preterm birth,[2 3] stillbirth,[4] diabetes in pregnancy[5] and pregnancy hypertension.[6] Although these processes have been invaluable for prioritising research themes, the broad topics commonly identified have not been readily amenable to testing in high-quality interventional studies. Methodologically robust randomised controlled trials (RCTs) are the gold standard for assessing effectiveness of a healthcare intervention, drug or technology[7] and are critical to improving quality and reducing variation in neonatal

**Example PICO from PlaNeT-2 Trial (Platelet transfusion thresholds in premature neonates)**[18]

**Participants:** Preterm Infants < 34 weeks gestation at birth

**Intervention:** Low transfusion threshold (transfusing patient if platelet count < $25\times10^9$/L)

**Control:** High transfusion threshold (transfusing patient if platelet count <$50\times10^9$/L)

**Outcome:** Primary Outcomes were mortality or major bleeding within 28 days. (Secondary Outcomes were bronchopulmonary dysplasia, sepsis, retinopathy of prematurity and necrotising enterocolitis.)

**Figure 1** Example PICO from PlaNeT-2 (Platelets for Neonatal Transfusion - 2) trial. PICO, Population, Intervention, Comparison, Outcome.

| Steering Group Member | Role and affiliation |
|---|---|
| Cheryl Battersby (CB) | Academic Neonatologist, BAPM Data/Informatics lead & member of NIHR prioritisation committee. |
| James Boardman (JB) | Professor of Neonatal Medicine and immediate past president of the Neonatal Society. |
| Elaine Boyle (EB) | Professor of Neonatal Medicine and Chair of the National Institute for Health Research Neonatal Clinical Studies Group. |
| William Carroll (WC) | Consultant Paediatrician and RCPCH officer for Research. |
| Jon Dorling (JD) | Professor of Paediatrics, Neonatal Consultant and BAPM research lead. |
| Kate Dinwiddy (KD) | Chief Executive of BAPM. |
| Katie Evans (KE) | Project Co-ordinator and Honorary Clinical Research Fellow in Neonatal Medicine. |
| Chris Gale (CG) | Academic Neonatologist and Neonatal Society Meeting Secretary. |
| Katie Gallagher (KG) | Academic Neonatal Nurse and Neonatal Nurses Association representative. |
| Pollyanna Hardy (PH) | Clinical Trials Statistician and Director of National Perinatal Epidemiology Unit. |
| Emma Johnston (EJ) | Parent representative and Parents and Family engagement Lead with the Thames Valley and Wessex ODN. |
| Helen Mactier (HM) | Consultant Neonatologist, Honorary Clinical Associate Professor and President of BAPM. |
| Claire Marcroft (CM) | Neonatal Physiotherapist and Allied Health Professionals Representative. |
| James Webbe (JW) | Trainee representative and Neonatal Medicine GRID Trainee. |

**Figure 2** Steering group members. BAPM, British Association of Perinatal Medicine; NIHR, National Institute for Health Research; RCPCH, Royal College of Paediatrics and Child Health; ODN, Operational Delivery Network.

care. Interventional studies such as RCTs require clearly structured and focused research questions, which can be organised using the PICO model, which guides the questioner to clearly identify the Participant (P), Intervention (I), Comparator (C) and Outcome (O)[8 9] (figure 1).[10]

There is wide variation in neonatal care[11 12] and an incomplete evidence base for many neonatal treatments.[13] Consequently, there is a need to identify and prioritise research questions that can be tested in randomised trials. Involving all relevant stakeholders in such a process will ensure that research addresses questions that are important to healthcare professionals, former neonatal patients and parents, as well as relevant to current neonatal care.

The aim of this project is to identify and prioritise neonatal research questions in PICO format, suitable for high-quality interventional studies, using a robust, transparent and inclusive methodology. The results of this identification and prioritisation process will inform the development and design of neonatal interventional studies in high-income settings, including the National Healthcare System in the United Kingdom.

## METHODS AND ANALYSIS
### Steering group
A steering group was formed in September 2021 to agree the scope and facilitate the process of identifying and prioritising research questions. The following key stakeholders are represented on the steering group: the Neonatal Society (NS), British Association of Perinatal Medicine (BAPM), Neonatal Nurses Association (NA), Royal College of Paediatrics and Child Health, neonatal allied health professionals (AHPs), neonatal clinical trial methodologists, neonatal trainees and parents with experience of neonatal care. The steering group is cochaired

by representatives of the NS and BAPM and has an allocated project co-ordinator responsible for agenda, minuting, role allocation and participant support. Details of the steering group members and their affiliations are found in figure 2.

The roles of the steering group are as follows:
► Agreeing the scope of the process.
► Disseminating details of the process.
► Engaging with clinical stakeholders to take part in identifying and prioritising research questions.
► Review of submitted research questions to identify duplicate questions, questions that have already been answered and questions outside the scope.
► Disseminating the final ranked list of research questions.

### Identification of stakeholders
The utility and validity of this project will depend on ensuring that representative questions are generated and prioritised by a wide group of neonatal stakeholders including:
► Clinicians involved in neonatal care: neonatologists, paediatricians, neonatal nurses, advanced neonatal nurse practitioners (ANNPs) working in paediatrics or neonatal medicine. Recruitment will be through advertisements on the RCPCH website and through other relevant professional organisations including BAPM, the Neonatal Nurses Association and the

NS. Trainee doctors and nurses will be additionally contacted through local training schools and use of communication methods such as regional teaching and social media channels.

► AHPs: occupational therapists, physiotherapists, dieticians, speech and language therapists and clinical psychologists working in neonatal care: advertisements will be placed through professional websites and organisations and co-ordinated by the AHP steering group representative.

► Academics and researchers working within neonatology: recruitment will be targeted through academic organisations, existing research networks, national meetings and through clinical trial units with a neonatal interest.

► Parents and former neonatal patients: these groups will be contacted through the national care coordinator group, maternity voices partnerships, charity websites and through social media platforms.

To ensure maximal engagement across all stakeholder groups, various communication strategies will be employing including email contact, professional websites and social media routes.

### Patient and public involvement

Parents and patients have been involved in this work from its inception. A parent representative sits on the steering group to plan the project and optimise parent and patient involvement. Communication to parents and former neonatal patients will be through parent and patient network charities, regional support networks and social media groups. We recognise the importance of hearing from minority ethnic communities and those from low socioeconomic backgrounds, particularly in view of their greater risks of pregnancy complications that may lead to experience of neonatal care.[14] These stakeholders were contacted directly through relevant parent-led and charitable organisations with requests to share details of the project on their social media platforms and email lists. The communication strategy prioritises working with a wide range of people from all backgrounds and plans to ensure accessible language translations of research questions during the final parental and former patient prioritisation process.

Parents and former neonatal patients will be involved in identifying neonatal research questions through the online identification process. We acknowledge that there will need to be extensive support for non-academic participants to ensure they are not dissuaded by the PICO structure. To further facilitate the identification of research questions from parents and former neonatal patients, we will develop targeted online support resources and run several online workshops focused on developing questions using a PICO format. We will develop a training video targeted specifically towards parents and former patients and develop our resource page on the BAPM website with further written information about formatting a PICO question. In addition, our question building software will be designed specifically to walk participants through the PICO process, with preformatted options for populations and outcomes to make it more accessible.

Parents and former neonatal patients will then be invited to participate in the prioritisation of research questions through the eDelphi process. Due to the nature of the study, some questions may require a high level of medical understanding, and so there will be the option for 'unable to rank' if a participant feels that they cannot make a meaningful judgement on assessing its importance. Throughout the eDelphi, we will seek feedback from parents and former patients regarding its accessibility. Balancing meaningful parental involvement with the specificity required for questions to be answered in interventional trials may be challenging. Therefore, with the results of the final eDelphi survey, we will carry out a specific targeted prioritisation process with parents, families and former patients, with support from Bliss, to ensure that this important group of voices is heard.

### Scope

The scope of the prioritisation process has been agreed by the steering group as follows:

► Limited to research questions relevant to high-income neonatal care settings.

► Limited to research questions related to care provided by neonatal teams:
  – On neonatal units, transitional care units or as part of neonatal transport.
  – On postnatal wards (excluding care exclusively provided by midwifery teams without neonatal input).
  – In the community after receiving neonatal care as an inpatient (care may be provided by community neonatal teams or associated community AHPs with neonatal interest).

### Overview

This study will be divided into four stages, which will commence in January 2022 and complete by September 2022 (figure 3):

1. Identification of neonatal research questions suitable for analysis in RCTs (1 month).
2. Review of submitted neonatal research questions to remove duplicate questions and previously answered questions (1 month).
3. Prioritisation of neonatal research questions by all relevant stakeholders using a three-round eDelphi process (3 months).
4. Dissemination of ranked list of PICO questions.

### Stage 1: identification of testable neonatal research questions

We will identify neonatal research questions suitable for evaluation in practice-changing RCTs through an open process. Individual stakeholders (neonatal clinicians, neonatal nurses, ANNPs, neonatal AHPs, neonatal researchers and former neonatal patients and patients) will be contacted via professional organisations, social media platforms, networks and organisational mailing

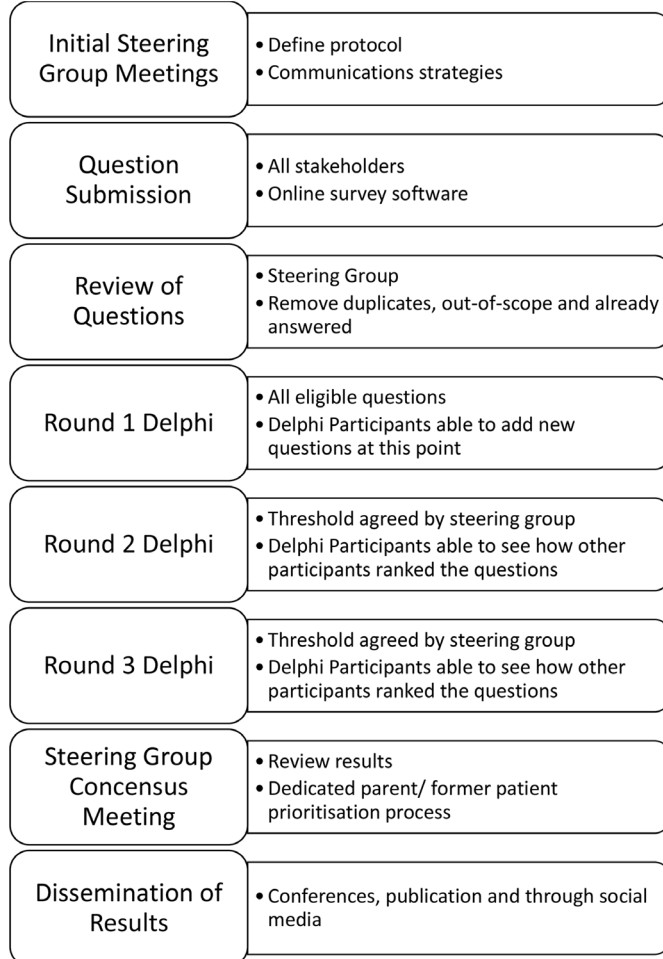

| Initial Steering Group Meetings | • Define protocol<br>• Communications strategies |
| --- | --- |
| Question Submission | • All stakeholders<br>• Online survey software |
| Review of Questions | • Steering Group<br>• Remove duplicates, out-of-scope and already answered |
| Round 1 Delphi | • All eligible questions<br>• Delphi Participants able to add new questions at this point |
| Round 2 Delphi | • Threshold agreed by steering group<br>• Delphi Participants able to see how other participants ranked the questions |
| Round 3 Delphi | • Threshold agreed by steering group<br>• Delphi Participants able to see how other participants ranked the questions |
| Steering Group Concensus Meeting | • Review results<br>• Dedicated parent/ former patient prioritisation process |
| Dissemination of Results | • Conferences, publication and through social media |

**Figure 3** Methods.

lists. The process for identifying research questions will also be openly publicised on organisational websites seeking submission of research questions from stakeholders. Publicity will precede and continue throughout a 4-week submission period[15] to optimise engagement and inclusion of all key groups.

Stakeholders will be invited to submit as many questions as they would like using an online system that facilitates submission using the PICO structure:

► Populations will be able to be selected from a predefined selection of gestations and clinical cohorts or specified by the submitter.
► Intervention and comparisons will take the form of free text to allow full descriptions.
► Outcomes will be able to be selected from Core Outcomes in Neonatology,[16] or specified by the submitter to ensure both core outcomes and individual outcomes are valued equally.

Support in structuring research questions using PICO will be facilitated by the steering group members and organisations using techniques such as short videos and webinars. Contact details and basic demographic data will be requested from individuals who submit questions to monitor the representativeness of stakeholder

involvement and to invite participation in the subsequent prioritisation process.

Submission of relevant PICOs from previous priority setting work[17] will be welcomed.

## Stage 2: review and refinement of the long list of neonatal research questions

Submitted research questions will be reviewed by the steering group to remove duplicate, already answered and out of scope questions, and to refine those not consistent with the PICO structure. Each question will be independently reviewed by two separate members of the steering group to ensure a transparent and reproducible process. It is recognised that there can be widely variable views on whether a clinical question has been adequately 'answered' by existing research and in view of this any questions excluded on this rationale will have to receive consensus view from all members of the steering group. Prior to exclusion, questions will be independently reviewed by two members of the steering group. A final long list of research questions will be taken forward for prioritisation.

## Stage 3: prioritisation of neonatal research questions

All research questions included in the long list will be entered into a three-round eDelphi prioritisation process. Involvement of stakeholders in the prioritisation process will be facilitated as follows:

► All stakeholders who submitted questions will be contacted by email and asked to take part.
► Invitations to take part will be circulated by professional organisations and open links will be made available on professional organisational websites and social media accounts.
► Parents and previous neonatal patients will be contacted through professional and charitable organisations and networks and through social media.

All research questions will be ranked using a 9-point Likert scale. After the first round, there will be the opportunity to submit new PICO questions. These additional questions will pass through the same two stage review process, co-ordinated by the steering group as the original questions. Prior to opening the second round of the Delphi survey, a steering group meeting will define the threshold for questions to take part in this round. During the second round, participants will be provided with information on how individual research questions were prioritised by each stakeholder group during the first round. Participants will then be able to amend or confirm their original ratings, taking into consideration other stakeholder group opinions. Prior to the third round, the steering group will convene a meeting and review the preliminary results to decide the threshold of questions to enter the third round. A shorter list of questions in the third round will help to improve retention of participants by facilitating a shorter time commitment.

After completion of the third round of ranking, all research questions entered into the Delphi will be collated

into a ranked list of research priorities for the UK neonatal community. Ranking will be ordered based on mean score. Results will also be presented by mean score within each stakeholder group. All questions not excluded in the initial review process will be included in the final list.

## ETHICS AND DISSEMINATION

Research Ethics Committee approval is not required for this work.

The ranked list of defined neonatal research questions will be disseminated as follows:

► We will share results directly with the NIHR prioritisation panel and other funders of neonatal trials.
► We will circulate the list to stakeholder organisations and their members.
► Individuals who participated in question setting or the prioritisation process will be emailed directly with the finalised rankings and named as group authors in any published work.
► Results will be disseminated among the scientific community through a publication and presented at relevant neonatal meetings.

## DISCUSSION

More than 1 in 10 babies in many high-income settings will receive neonatal care. Neonatal conditions are important, contributing to almost half of all child deaths in the UK and to many long-term health conditions. Despite this importance, much neonatal care is not based on high-quality research evidence. Historically, the perinatal research environment has fostered collaborative working, demonstrated by studies such as EPICure.[18] The SARS-CoV-2 pandemic has further highlighted the benefits of joined up working across multiple geographical regions, scientific institutions and research groups in achieving practice changing outcomes within much shorter timeframes.[19] This has resulted in increased focus on prioritisation projects within healthcare using initiatives such as the James Lind Alliance to support unified decision-making across specific specialities. Neonatology within the UK is ideally placed for such prioritisation work and future collaborative research due to centrally funded healthcare, the neonatal operational delivery network structure facilitating close relationships between units and the accessibility of large national databases. Ultimately, this process seeks to involve a wide range of key neonatal stakeholders, to identify and prioritise research questions addressing the many clinical uncertainties, suitable for evaluating in well-structured clinical trials.

## Author affiliations
[1]Neonatal Medicine, School of Public Health, Faculty of Medicine, Imperial College London, London, UK
[2]MRC Centre for Reproductive Health, The University of Edinburgh, Edinburgh, UK
[3]Neonatal Medicine, Department of Health Sciences, University of Leicester, Leicester, UK
[4]Department of Paediatric Respiratory Medicine, University Hospitals of North Midlands, Stoke-on-Trent, UK
[5]Chief Executive of British Association of Perinatal Medicine, London, UK
[6]Neonatal Medicine, University Hospital Southampton, Southampton, UK
[7]Child and Adolescent Health, University College London, EGA Institute for Women's Health, London, UK
[8]National Perinatal Epidemiology Unit, Clinical Trials Unit, University of Oxford, Oxford, UK
[9]Parents and Families Engagement Lead, Thames Valley and Wessex Operational Deliveries Network, Thames Valley and Wessex, UK
[10]Neonatal Medicine, University of Glasgow, Glasgow, UK
[11]Population Health Sciences Institute, Faculty of Medical Sciences, Newcastle University, Newcastle upon Tyne, UK

**Acknowledgements** We wish to thank the members of the Neonatal Priority Setting Partnership Steering Group for their support and RCPCH, BAPM, NNA and the NS for their support.

**Contributors** CG and CB conceived this project. KE, JPB, HM, CG and CB planned and coordinated the initial steering group and protocol. KD provided administrative support. The first draft of the manuscript was written by KE and revised by CB, JPB, EB, WDC, JD, KG, PH, EJ, HM, JW and CG. CG edited and reviewed the manuscript. It was approved by all members of the steering group.

**Funding** Administrative support for the project is provided by BAPM; questionnaire and online Delphi software are funded by the Medical Research Council (MRC) through a Transition Support Award held by CG (MR/V036866/1).

**Competing interests** CG is vice chair of the NIHR Research for Patient Benefit London Regional Advisory Panel and a member of the Glasgow Children's Hospital Charity External Panel; he holds a Medical Research Council Transition Support Award. JPB is a member of the Wellcome Trust's Career Development Award Panel and the Great Ormond Street Hospital Charity Research Assessment Panel. CB is the NIHR deputy chair of HTA prioritisation committee for hospitals. JD is a member of the NIHR HTA CET Funding Committee. CM is funded by HEE-NIHR Integrated Clinical Academic Programme and holds a NIHR ICA CSRF Fellowship.

**Patient and public involvement** Patients and/or the public were involved in the design, or conduct, or reporting, or dissemination plans of this research. Refer to the Methods section for further details.

**Patient consent for publication** Not applicable.

**Provenance and peer review** Not commissioned; externally peer reviewed.

**ORCID iDs**
Katie Evans http://orcid.org/0000-0002-0503-9807
Cheryl Battersby http://orcid.org/0000-0002-2898-553X
James P Boardman http://orcid.org/0000-0003-3904-8960
Katie Gallagher http://orcid.org/0000-0002-6847-9594
Helen Mactier http://orcid.org/0000-0001-6154-5758
James Webbe http://orcid.org/0000-0001-8546-3212
Chris Gale http://orcid.org/0000-0003-0707-876X

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
