## [Reviewer comments · BMJ Open]

ARTICLE DETAILS

TITLE (PROVISIONAL)	Protocol: A national priority setting partnership using a Delphi consensus process to develop neonatal research questions suitable for practice-changing randomised trials in the United Kingdom
AUTHORS	Evans, Katie; Battersby, Cheryl; Boardman, James; Boyle, Elaine; Carroll, William; Dinwiddy, Kate; Dorling, Jon; Gallagher, Katie; Hardy, Pollyanna; Johnston, Emma; Mactier, Helen; Marcroft, Claire; Webbe, James; Gale, Chris

VERSION 1 – REVIEW

REVIEWER	Wielenga, Joke M. Curtin Univ
REVIEW RETURNED	15-Mar-2022

GENERAL COMMENTS	Clearly important work is started with this priority setting partnership, with all stakeholders, to develop neonatal research for practice changing. it is clearly stated that the scope is narrowed by topics that are suitable for RCT methodology. Practice changing needs more than research questions answered by RCT's and I hope the next step will be to prioritise those as well. A few comments: 1. Dates of the study are lacking, also a timeframe is lacking;2. The methods are not yet detailed enough, for me it is rough description of plans considering this is a protocol;3.The Outcomes from the Core Outcomes in Neonatology (COIN) are very medically orientated, except QoL. I would like to see nursing, social and parenteral outcome to be able to selected instead of added by the submitter. To state that these are equally important, adding by the summitters suggests a differentiation in importance;4. What is asked of the participants in the second Delphi round?
---

REVIEWER	Heazell, Alexander University of Manchester, Maternal and Fetal Health Research Centre
REVIEW RETURNED	13-May-2022

GENERAL COMMENTS	The authorship team is multidisciplinary and includes members from relevant stakeholder organisations and a lay member. Would it be appropriate to access additional stakeholder support from organisations such as BLISS? I understand the authors desire to want to prioritise topics for randomised controlled trials, but these are not the only methodological approach that can be applied. Whilst these remove selection bias they are subject to other issues including
---

	participation bias. The size of RCTs is also problematic for rare outcomes e.g. neonatal mortality. In the introduction I wonder whether the authors would consider replacing randomised trials with high-quality intervention studies, as this would recognise that while RCTs might be regarded as the gold standard these might not always be feasible or generalisable. It would be good if the authors can clarify how the steering group is chaired. Will this be chaired by a lay member, clinician or academic? I note that the authors are not working with the James Lind Alliance which means they do not have an independent facilitator which can be a strength. It would be good to be clearer about the organisational structure. There is a danger that requiring responses in a PICO format could dissuade non-academic participants. I thought the inclusion of short training videos was a good idea. I think some additional written information might also be helpful. It wasn't clear how the authors will reach seldom heard voices such as those from the Black or Asian Communities or deprived communities (all of which have greater risk of pregnancy complications that may necessitate neonatal care). My experience is that this requires focussed efforts from the research team liaising with specific community organisations. I would strongly encourage the authors to include this in their protocol. How long do the authors think that the individual phases will last? It would be helpful to give some indication. In stage 2 how will the authors know that questions have already been answered? How will the literature searches be conducted (in what databases and by whom), what grade of evidence will they require to determine whether a question has been answered? Who will make that decision? The protocol would benefit from more detail here.
--	---

VERSION 1 – AUTHOR RESPONSE

Reviewer: 1

Dr. Joke M. Wielenga, Curtin Univ

Comments to the Author:

Clearly important work is started with this priority setting partnership, with all stakeholders, to develop neonatal research for practice changing. It is clearly stated that the scope is narrowed by topics that are suitable for RCT methodology. Practice changing needs more than research questions answered by RCT's and I hope the next step will be to prioritise those as well.

Many thanks for your comments and whilst this initiative is focussed on prioritisation of research questions suitable for high-quality intervention trials, we fully recognise that other research modalities are highly valuable for progressing clinical practice. Our steering group has discussed understanding similar prioritisation processes in the future looking at other research methodologies and quality improvement, however this was deemed outside of the scope of this current project. We hope that this project will lay the ground for further priority setting work, looking at other research designs.

1. Dates of the study are lacking, also a timeframe is lacking;

Additional dates and timeframe have been added to the Methods section:

- Steering Group: "was formed in September 2021".
- Overview: "which will run from January 2022 and complete by September 2022"

2. The methods are not yet detailed enough, for me it is rough description of plans considering this is a protocol;

Additional information added throughout Methods section:

- Figure 3: Clear diagram showing the different stages of the project.
- Stage 1: "and webinars"
- Stage 3: "These additional questions will pass through the same two stage review process coordinated by the steering group as the original questions."
- Stage 3: Additional information regarding the thresholds for question inclusion in each stage of the Delphi survey. "Prior to opening the second round of the Delphi survey, a steering group meeting will define the threshold for questions to take part in this round"

3. The Outcomes from the Core Outcomes in Neonatology (COIN) are very medically orientated, except QoL. I would like to see nursing, social and parental outcome to be able to selected instead of added by the submitter. To state that these are equally important, adding by the sumitters suggests a differentiation in importance;

Many thanks for your helpful comment. The Core Outcomes in Neonatology project involved the input of over 200 former neonatal patients and parents who rated many outcomes including nursing, social and parental outcomes. The final outcomes were those agreed by all stakeholders to be crucially important, and so we have used them as a starting point. The use of the Core Outcomes in Neonatology was discussed and agreed on in our steering group meetings, in view of the fact that they reflected the important outcomes selected by a wide range of neonatal stakeholders, particularly parents. We agree that other outcomes are often more important for particular research questions, and thus we have ensured careful phrasing on the question submission software, to emphasise that both core outcomes and individual outcomes are important, therefore hope that this will not benterpreted as a difference in importance.

- We have added this to the Methods "Outcomes will be able to be selected from Core Outcomes in Neonatology (COIN) or specified by the submitter to ensure both core outcomes and individual outcomes are valued equally".

4. What is asked of the participants in the second Delphi round?

Additional information added in Methods Section:

- "Participants will be provided with information on how individual research questions were prioritised by each stakeholder group during the first stage"
- "Participants will then be able to amend or confirm their original ratings taking into consideration the views of other stakeholder group opinions."
- Figure 3 added to show each stage of the process in detail.

Reviewer: 2

Dr. Alexander Heazell, University of Manchester

Comments to the Author:

1. The authorship team is multidisciplinary and includes members from relevant stakeholder organisations and a lay member. Would it be appropriate to access additional stakeholder support from organisations such as BLISS?

Many thanks for your helpful comments. We have liaised with a number of stakeholder organisations, including BLISS throughout the conception of this project and have grateful received feedback and support in aiming to ensure meaningful participation for parents, families and former patients. We have updated the PPI section of thprotocol with additional details of how we aim to work with multiple stakeholder organisations to try and ensure broad neonatal community representation throughout this project. We will undertake a parent and former patient focussed prioritisation process of the top ranked questions at the end of the eDelphi in association with Bliss and have updated the methods section accordingly.

2. I understand the authors desire to want to prioritise topics for randomised controlled trials, but these are not the only methodological approach that can be applied. Whilst these remove selection bias they are subject to other issues including participation bias. The size of RCTs is also problematic for rare outcomes e.g. neonatal mortality. In the introduction I wonder whether the authors would consider replacing randomised trials with high-quality intervention studies, as this would recognise that while RCTs might be regarded as the gold standard these might not always be feasible or generalisable.

Many thanks for this comment. The best way of phrasing our aims was debated extensively in early steering group meetings, as we agree completely that RCTs are by no means the only important research design. We hope that this project will lay the groundwork for future priority setting work which may focus on other important research methods and designs. We gratefully accept your comment regarding using the term “intervention study” and have rephrased this throughout the introduction.

3. It would be good if the authors can clarify how the steering group is chaired. Will this be chaired by a lay member, clinician or academic? I note that the authors are not working with the James Lind Alliance which means they do not have an independent facilitator which can be a strength. It would be good to be clearer about the organisational structure.

The steering group is co-chaired by representatives of the Neonatal Society (Dr Chris Gale – Academic Neonatologist and Dr Helen Mactier - BAPM President). Ms Kate Dinwiddy (BAPM Chief Executive) sits on the group as a lay person and provides administrative support. Dr Katie Evans acts as the project co-ordinator, responsible for agenda setting, minutes and co-ordination of all aspects of the initiative such as the review stages. We are grateful for your comments and have clarified the organisation structure further in the Methods: Steering Group section of the manuscript.

- “has an allocated project co-ordinator responsible for agenda setting, minuting, role allocation and participant support”

- “Details of the steering group members and their affiliations can be found in figure 2.”

4. There is a danger that requiring responses in a PICO format could dissuade non-academic participants. I thought the inclusion of short training videos was a good idea. I think some additional written information might also be helpful.

Many thanks for your comments and we acknowledged early on that there would need to be extensive support for non-academic participants to ensure they were not dissuaded by the PICO structure. However, the steering group also felt that the PICO structure was very important to ensure questions were detailed and balanced enough for answering in intervention studies. We have produced several training videos targeted towards different stakeholder groups, alongside two separate BAPM webinars – one designed specifically to support parents. There is a dedicated resource page on the BAPM website with further written information about how to format a PICO question. Additionally, our question building software was designed specifically to walk participants through the process – with pre-formatted options for populations and outcomes to make it more accessible. We have updated the methods and PPI sections with this additional information.

5. It wasn't clear how the authors will reach seldom heard voices such as those from the Black or Asian Communities or deprived communities (all of which have greater risk of pregnancy complications that may necessitate neonatal care). My experience is that this requires focussed efforts from the research team liaising with specific community organisations. I would strongly encourage the authors to include this in their protocol.

Many thanks for this helpful comment and we have highlighted this in the Patient and Public Involvement section of the protocol.

- “We recognise the importance of hearing from minority ethnic communities and those from low socio-economic backgrounds, in view of their greater risks of pregnancy complications that may lead

to experience of neonatal care. These stakeholders were contacted directly through relevant parent-led and charitable organisations with requests to share details of the project on their social media platforms and email lists. The communications strategy prioritises working with a wide range of people from all backgrounds, and plans to ensure accessible language translations of research questions during the final parental and former patient prioritisation process.”

- “Throughout the eDelphi we will seek feedback from parents and former patients regarding its accessibility. Balancing meaningful parental involvement with the specificity required for questions to be answered in interventional trials may be challenging. Therefore after the eDelphi completed we will facilitate a specific targeted prioritisation process with parents, families and former patients, with support from Bliss, to ensure that this important group of voices are heard.”

6. How long do the authors think that the individual phases will last? It would be helpful to give some indication.

We had added additional timeline information in the Overview section.

- “which will commence in January 2022 and complete by September 2022”
- Phase 1 “1 month”, Phase 2 “1 month”, Phase 3 “3 months”

7. In stage 2 how will the authors know that questions have already been answered? How will the literature searches be conducted (in what databases and by whom), what grade of evidence will they require to determine whether a question has been answered? Who will make that decision? The protocol would benefit from more detail here.

We have added further detail about stage 2 and the review process in the body of the methods.

- “Each question will be independently reviewed by two separate members of the steering group to ensure a transparent and reproducible process.”
- “It is recognised that there can be widely variable views on whether a clinical question has been adequately ‘answered’ by existing research and in view of this any questions put forward for exclusion on this rationale will have to receive consensus view from all members of the steering group.”

Reviewer: 1

Competing interests of Reviewer: None

Reviewer: 2

Competing interests of Reviewer: I do not have any competing interests to declare.

VERSION 2 – REVIEW

REVIEWER	Heazell, Alexander University of Manchester, Maternal and Fetal Health Research Centre
REVIEW RETURNED	29-Aug-2022
GENERAL COMMENTS	I apologize for the delay in my review of these minor revisions. I am completely happy that the authors have addressed my comments.